# Predictors of Mortality in Patients with Advanced Cancer—A Systematic Review and Meta-Analysis

**DOI:** 10.3390/cancers14020328

**Published:** 2022-01-11

**Authors:** Catherine Owusuaa, Simone A. Dijkland, Daan Nieboer, Agnes van der Heide, Carin C. D. van der Rijt

**Affiliations:** 1Department of Medical Oncology, Erasmus MC Cancer Institute, P.O. Box 2040, 3000 CA Rotterdam, The Netherlands; c.vanderrijt@erasmusmc.nl; 2Department of Public Health, Erasmus MC, Erasmus University Medical Center, P.O. Box 2040, 3000 CA Rotterdam, The Netherlands; s.dijkland@erasmusmc.nl (S.A.D.); d.nieboer@erasmusmc.nl (D.N.); a.vanderheide@erasmusmc.nl (A.v.d.H.)

**Keywords:** advance care planning, predictors, mortality, advanced cancer

## Abstract

**Simple Summary:**

This systematic review and meta-analysis describes the predictors of mortality in patients with advanced cancer. The results indicate that disease stage, lung cancer, ECOG performance status, age, male sex, Charlson comorbidity score, and other multicomponent prognostic models could aid physicians in timely advance care planning. However, combining these predictors in a prognostic model with adequate performance requires more research.

**Abstract:**

To timely initiate advance care planning in patients with advanced cancer, physicians should identify patients with limited life expectancy. We aimed to identify predictors of mortality. To identify the relevant literature, we searched Embase, MEDLINE, Cochrane Central, Web of Science, and PubMed databases between January 2000–April 2020. Identified studies were assessed on risk-of-bias with a modified QUIPS tool. The main outcomes were predictors and prediction models of mortality within a period of 3–24 months. We included predictors that were studied in ≥2 cancer types in a meta-analysis using a fixed or random-effects model and summarized the discriminative ability of models. We included 68 studies (ranging from 42 to 66,112 patients), of which 24 were low risk-of-bias, and 39 were included in the meta-analysis. Using a fixed-effects model, the predictors of mortality were: the surprise question, performance status, cognitive impairment, (sub)cutaneous metastases, body mass index, comorbidity, serum albumin, and hemoglobin. Using a random-effects model, predictors were: disease stage IV (hazard ratio [HR] 7.58; 95% confidence interval [CI] 4.00–14.36), lung cancer (HR 2.51; 95% CI 1.24–5.06), ECOG performance status 1+ (HR 2.03; 95% CI 1.44–2.86) and 2+ (HR 4.06; 95% CI 2.36–6.98), age (HR 1.20; 95% CI 1.05–1.38), male sex (HR 1.24; 95% CI 1.14–1.36), and Charlson comorbidity score 3+ (HR 1.60; 95% CI 1.11–2.32). Thirteen studies reported on prediction models consisting of different sets of predictors with mostly moderate discriminative ability. To conclude, we identified reasonably accurate non-tumor specific predictors of mortality. Those predictors could guide in developing a more accurate prediction model and in selecting patients for advance care planning.

## 1. Introduction

Advance care planning is important in patients with cancer because it enables physicians and patients to discuss and make plans for future health care [1]. It should be initiated timely in patients with advanced cancer, as a decline in the functional status of the patient, although anticipated, often occurs quickly [2]. Therefore, it is a prerequisite for the responsible physician to have insight into the limited life expectancy of their patients. 

It is often suggested that patients who are likely to have a maximum life expectancy of 12 months are candidates for advance care planning. Therefore, a period of 12 months before death may be typical for patients with palliative care needs and has been included as such in quality standards worldwide [3,4]. The identification of such patients can be based on the surprise question, ‘Would you be surprised if this patient died within the next 12 months?’ [5]. Although the surprise question was initially developed to encourage physicians to refer their patients for an advance care planning program, it has also been studied as a prediction model for 12-month mortality [6]. However, it has an accuracy ranging from poor to reasonable, with a pooled accuracy level of 78.6% (95% confidence interval 69.7–86.3) for patients with cancer [7]. Other prediction models such as the Palliative Prognostic Index, the Palliative Prognostic Score, and the Palliative Performance Scale focus on a life expectancy of only weeks or a few months, which is probably too short for adequate advance care planning procedures [8]. 

Better insight into whether patients are likely to die within a year might aid physicians in the timely initiation of advance care planning in all patients with advanced cancer who qualify for and would benefit from it. We aimed to provide an overview of predictors and prediction models for mortality within a period of 3–24 months for patients with advanced cancer. By examining all cancer types, we intended to identify general predictors that can be applied by various caregivers.

## 2. Materials and Methods

### 2.1. Search Strategy and Selection Criteria

The search strategy, which included the terms “advanced cancer”, “mortality”, “death”, “predictor”, and “prediction model”, was developed by one researcher (CO) and an information specialist at the Erasmus MC Medical Library. They performed the search in April 2020 in the following online databases: Embase, MEDLINE, Cochrane Central, Web of Science, and PubMed (Appendix A). An online study protocol for this systematic review and meta-analysis (including a search for chronic pulmonary diseases) was published at PROSPERO with registration number CRD42016038494; link: https://www.crd.york.ac.uk/PROSPERO/display_record.php?RecordID=38494 (accessed on 6 January 2022). 

Studies included in this systematic review and meta-analysis had a population with (predominantly) locally advanced or metastatic cancer; studied mortality within a period of 3–24 months or found a survival between 3–24 months; reported on risk estimates (hazard ratio, odds ratio or relative risk) with their corresponding standard errors or reported on the performance of a prediction model (discriminative ability: area under the receiver operating characteristic curve [AUC], c-statistic, sensitivity, or specificity); and were published in English and from the year 2000 onwards. Both cohort studies and randomized controlled trials were assessed. We excluded systematic reviews and studies that examined mortality or survival outside the period of 3–24 months; only included patients who were treated with curative intent; had a combined outcome (e.g., mortality or hospitalization); or did not report standard errors.

The titles of papers that resulted from the search were downloaded into a reference management software program (EndNote, version X9) and screened by C.O. [9]. Afterwards, C.O. and S.A.D. independently reviewed abstracts and full-text articles to determine the studies’ inclusion. If they disagreed, the full article was reassessed, and if necessary, two other researchers (A.v.d.H. and C.C.D.v.d.R.) were involved. When duplicate study data were found in different papers, only the most recent publication was included. We requested full articles that were unavailable to us from the first author through email. We also performed a grey literature search using the bibliographies from included studies and from systematic reviews for additional relevant studies.

### 2.2. Data Extraction

C.O. and S.A.D. independently extracted the following data from each included study: first author, publication year, study design, study population (number of patients, age, sex, and cancer type), follow-up period, mortality rate, overall survival, risk estimates and standard errors, and the predictors and discriminative ability of a prediction model. Means were derived with standard deviations and medians with (interquartile) range. When studies reported on different follow-up periods, we only extracted the data of the longest follow-up period within the inclusion criteria. All extracted data were entered in a specifically designed Microsoft Access database. 

C.O. and S.A.D. also independently extracted data to assess the risk-of-bias of individual studies using the Quality in Prognosis Studies with adapted items from the Critical Appraisal and Data Extraction for Systematic Reviews of Prediction Modelling Studies checklist [10,11]. A total of 23 items were divided over the following six domains: study participation, study attrition, predictor measurement, outcome measurement, statistical analysis and confounding, and model performance (Appendix A). All items were given 2 points if bias was not present, 1 point if bias was possibly present, and 0 points if bias was obviously present. The overall risk-of-bias (low [≥80% of total points], moderate [≥60–79% of total points], or high [<60% of total points]) of each study was generated from the points given to the items per domain. Disagreements in scoring were resolved by discussion between the researchers. This study was conducted according to the Preferred Reporting Items for Systematic and Meta-analyses (PRISMA) guideline [12].

### 2.3. Data Analysis

The main outcomes were predictors and prediction models for mortality within a period of 3–24 months in patients with advanced cancer. The risk estimates and standard errors of predictors that were retrieved from multivariable analyses in published studies were included in the meta-analysis, provided that they were studied in ≥2 (various) cancer types [13]. We calculated the pooled overall risk estimate and standard error with a fixed-effects model for predictors that were pooled from two studies and with a random-effects model for predictors pooled from three or more studies. The fixed-effects model was preferred over the random-effects model if two studies were available, as estimates of heterogeneity become unreliable in a meta-analysis with a few studies. When applying a random-effects model, we calculated the between-study heterogeneity with the I^2^–statistic [14]. A heterogeneity of 0–30% was interpreted as insignificant, 40–60% as moderate, and >60% as substantial. We also performed a meta-analysis with only low risk-of-bias studies applying a fixed or random-effects model where appropriate. For performance status, we mapped different cut-offs of the Karnofsky performance scale with the Eastern Cooperative Oncology Group (ECOG) performance status [15]. All meta-analyses were conducted with the Review Manager 5.3 software (Cochrane Collaboration’s Information Management System).

For the prediction models that were studied in ≥2 (various) cancer types, we summarized the predictors and model performances. The discriminative ability measured with the AUC or c-statistic ranges from 0.5 to 1, and the closer the AUC or c-statistic is to 1, the better a model can discriminate patients who are likely to die from those who are not [16,17]. Furthermore, we identified publication bias using funnel plots [18].

## 3. Results

The literature search identified 6436 unique studies, of which 6265 were excluded based on their title and abstract (Figure 1). A total of 171 studies were screened for full text. Eventually, a total of 68 studies were systematically interpreted and summarized in the qualitative synthesis, whereupon 39 studies were included in the meta-analysis. 

Of the included studies, 63 were cohort studies, and five were analyses from randomized controlled trials. The number of patients ranged from 42 to 66,112 among all studies and from 42 to 13,190 among studies included in the meta-analysis (Table 1). Mean and median reported ages in all studies ranged from 54.0 and 79.4 years. Thirty-three studies had a study population with ≥2 cancer types, whilst the remaining studies included patients with one type of cancer. Mortality rates within the included studies ranged from 8.3 to 94.6%. Twenty-four studies were assessed as low risk-of-bias studies, 38 as moderate risk-of-bias, and six as high risk-of-bias (Table 2). Of the risk-of-bias domains, study attrition, which included items regarding loss to follow-up and handling of missing data, was often scored as high risk-of-bias.

In the fixed-effects model, all identified predictors of mortality were significantly associated with mortality (Figure 2). Those predictors were: the surprise question (HR 7.57; 95% CI 4.41–12.99), ECOG performance status 3–4 (reference [ref]: 0; HR 3.61; 95% CI 2.58–5.05), presence of (sub)cutaneous metastases (HR 2.10; 95% CI 1.51–2.93), and the Charlson co-morbidity index score 1–2 (HR 1.28; 95% CI 1.20–1.37). Body mass index, cognitive impairment (mini-mental state examination score), serum albumin, and hemoglobin were also significant predictors of mortality.

In the random-effects model, we included 14 predictors of mortality, of which 11 were significant (Figure 3 and Appendix A). Due to the low number of studies reporting on categorical variables, we performed a separate meta-analysis for each of the sub-categories. Significant predictors were: disease stage III (pooled from three studies; HR 2.89; 95% CI 1.42–5.90); disease stage IV (pooled from three studies; HR 7.58; 95% CI 4.00–14.36); lung cancer (pooled from four studies; HR 2.51; 95% CI 1.24–5.06). Age per 10-year increase, male sex, and a Charlson comorbidity index score of 3+ were also significant predictors, as were the Karnofsky performance status, ECOG performance status 1+ (ref: 0–1), 1–2 (ref: 0), 2 (ref: 0), and 2+ (ref: 0). Non-significant predictors were ECOG performance status 1 (ref: 0), ECOG performance status (ref: 0–1), and the presence of liver metastases. There was substantial between-study heterogeneity, i.e., I^2^ > 60%, for the age, ECOG performance status, comorbidity, lung cancer, and liver metastasis. Furthermore, no evidence of major publication bias was identified for each predictor (Appendix A).

In a separate analysis, we only included the predictors found in low risk-of-bias studies, namely age, male sex, and ECOG performance status 1+, of which only age and male sex were significantly associated with mortality (Appendix A). Variables that were excluded for the meta-analyses due to different cut-offs or definitions, or were studied in only one cancer type, are presented in Appendix A.

Thirteen studies reported on and analyzed the discriminative ability of 15 different prediction models, which consisted of one variable or combined different clinical variables such as cancer type, performance status, or metastases (Table 3). Most models had a moderate discriminative ability with a c-statistic or AUC ranging from 0.6–0.8, and others, such as the (Oncological-) multidimensional prognostic index, had a good discriminative ability of >0.8. The prediction models contained different sets of variables and were examined in just one study, except the surprise question, which was examined in two studies. The external validation of only two models was described.

## 4. Discussion

We aimed to identify significant predictors of mortality within a period 3–24 months for patients with advanced cancer in published literature. Overall, the predictors we found can be categorized into three main groups: the surprise question; clinical variables (age, male sex, disease stage, performance status, comorbidity, cognitive impairment, (sub)cutaneous metastases, and lung cancer); and laboratory variables (serum albumin and hemoglobin). Almost all predictors were non-tumor specific and may improve the identification of cancer patients at risk of death in the set time period. Better identification might enable conversations on advance care planning. However, the usefulness of some predictors in the clinical setting may be questionable. It might seem obvious that an advanced disease stage (III or IV) compared to a localized stage (I) is associated with mortality. Considering hemoglobin, it is unknown whether chemotherapy-induced or tumor-induced anemia were specifically considered, although this may be relevant. Furthermore, the identified predictors might not be exclusively associated with mortality within 3–24 months. For example, the performance status of a patient is, on the one hand, included in indexes to predict the terminal phase, such as in the Palliative Prognostic Score, the Palliative Prognostic Index, and the Chuang Prognostic Score [87]. On the other hand, the performance status is also included in tumor-specific indexes to predict mortality within a longer time frame exceeding 24 months [88,89,90].

The advantage of non-tumor specific predictors is that they apply to patients with various types of cancer. Furthermore, those predictors could be used by both medical specialists and other physicians, such as general practitioners. In theory, studies that were conducted in heterogeneous groups of patients, i.e., patients with various types of cancer, comorbidity and other characteristics, were most useful for our study. However, approximately half of the studies included patients with a specific type of cancer or a specific complication, as for example, bone metastases. To find non-tumor specific predictors in the meta-analysis, we only included predictors that were examined in heterogeneous study populations or in at least two groups of patients with different types of cancer. As a result, we observed substantial heterogeneity across studies in the meta-analysis for the predictors: age, the different variants of ECOG performance status, Charlson comorbidity index, lung cancer, and liver metastasis. However, the observed heterogeneity may also have been caused by different follow-up periods in the separate studies, different mortality rates, and differences between studies regarding the percentage of patients with locally advanced vs metastatic disease. Therefore, the substantial heterogeneity we found for the mentioned predictors can be expected. For example, the presence of liver metastasis in a patient with stomach cancer probably has a worse prognosis than in a patient with breast cancer. In addition, a patient with a poor performance status might seem to have a worse prognosis when examining a period of 24 months compared to a period of six months. Nevertheless, although heterogeneity was found between studies, we found various generic predictors of mortality. To optimize the prediction of mortality in a heterogeneous cancer population, multiple predictors must be considered together. We recommend the examination of several predictors in a prospective study in a heterogeneous cancer population.

Our study demonstrates the clinical importance of the surprise question, which was one of the most significant predictors in the meta-analysis. However, this finding should be interpreted with some caution because the surprise question was examined in only two studies. Furthermore, it is important to bear in mind that the surprise question has an aspect of subjectivity because it relates to the physician’s opinion [91]. Therefore, the generalizability of the surprise question might be questioned. Despite this, the surprise question is commonly used in the identification of patients with palliative care needs and included in palliative care needs assessment tools, such as the RADboud indicators for Palliative Care needs (RADPAC), Supportive and Palliative Care Indicators Tool (SPICT), and Prognostic Indicator Guide (PIG) [92]. Despite the fact that the surprise question is widely believed to be useful, there is abundant room for discussion concerning whether it should be used as a predictor of mortality, possibly combined with other variables, or only included in the assessment of palliative care needs [93,94].

Remarkably, apart from the surprise question, all other prediction models consisted of different variables or sets of variables. Their discriminative ability was mostly moderate, and some were good. However, a note of caution is due in the applicability of almost all the models since their calibration, which shows observed and predicted risk of mortality, was not studied. Another source of caution is the lack of external validation of the models. Despite the possible limitations of the models, two models with a good discriminative ability might be interesting for further study: the Patient-generated Subjective Global Assessment, which was externally validated; and the Oncological-multidimensional prognostic index, which had a good calibration. Nonetheless, there is a definite need for a general, high-quality prediction model for the accurate prediction of mortality in patients with advanced cancer.

This study had some limitations. Firstly, because we performed a meta-analysis, some predictors with slight differences in measuring units or cut-offs could not be pooled. An example of such a predictor was weight loss, which was studied at various cut-offs of kilograms. Therefore, the number of pooled studies for each predictor was small, e.g., the inclusion of two studies, which might limit the strength of a meta-analysis. Secondly, studies with moderate or high risk-of-bias scored poorly on the study attrition of our modified QUIPS tool. Lastly, the inclusion of published studies, exclusively from 2000 onward, could have led to the exclusion of some predictors of mortality.

## 5. Conclusions

The surprise question and general clinical (age, male sex, disease stage, performance status, comorbidity, cognitive impairment, (sub)cutaneous metastases, and lung cancer) and laboratory variables (serum albumin and hemoglobin) are non-tumor specific predictors of mortality within 3–24 months in patients with advanced cancer. Physicians could apply those predictors of mortality within 3–24 months in the timely initiation of advance care planning. Due to the lack of high-quality prediction models, the identified predictors can provide guidance in selecting candidate predictors for future prospective studies, which will explore the development of a general prediction model for mortality. Furthermore, future research should also examine the association of using such a prediction model with the timely initiation of advance care planning.

## Figures and Tables

**Figure 1 cancers-14-00328-f001:**
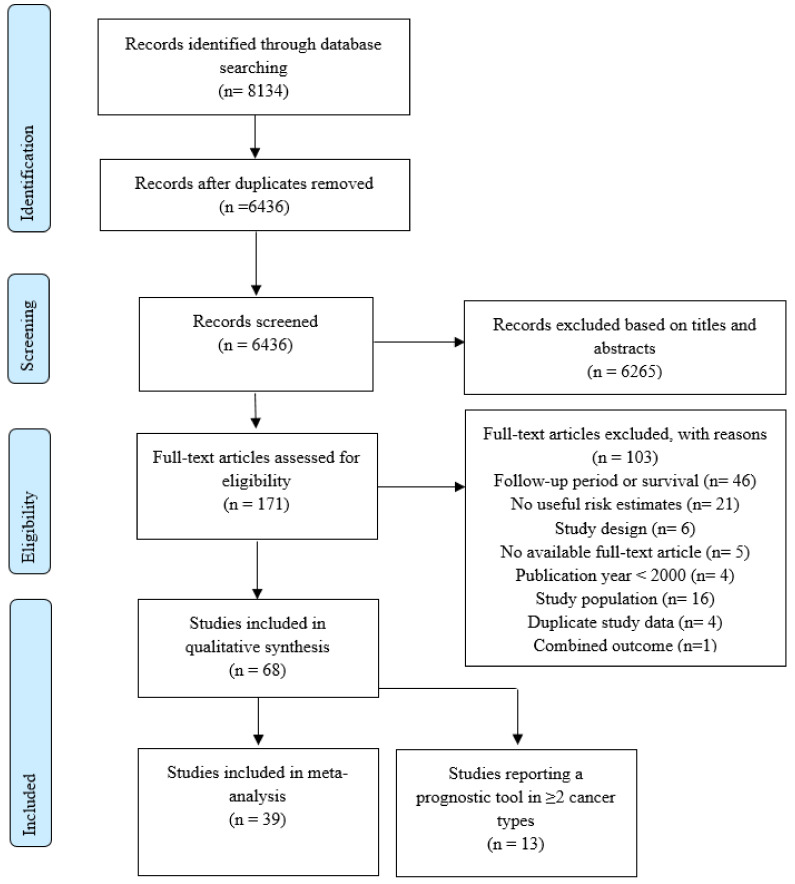
Study selection.

**Figure 2 cancers-14-00328-f002:**
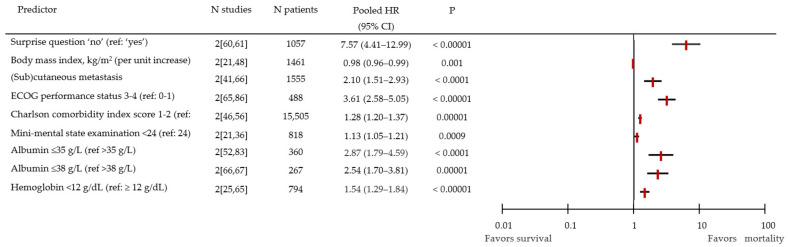
Forest plot of pooled hazard ratios for mortality with a fixed-effects model. CI: confidence interval; HR: hazard ratio; ECOG: Eastern Cooperative Oncology Group.

**Figure 3 cancers-14-00328-f003:**
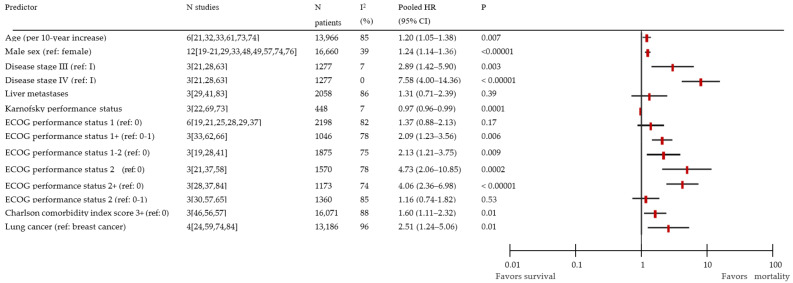
Forest plot of pooled hazard ratios for mortality with a random-effects model. CI: confidence interval; HR: hazard ratio; ECOG: Eastern Cooperative Oncology Group.

**Table 1 cancers-14-00328-t001:** Study characteristics of all 68 studies reporting predictors of mortality or prediction models.

	Study (First Author, Publication Year)	Study Design	N Patients	Study Population	Age (Years)	Men (%)	Follow-Up (Months)	Mortality Rate (%)	Survival (Months)	Inclusion in Meta-Analysis	Inclusion of a Model
1	Bartels, 2007 [19]	Cohort	219	Spinal epidural metastases; various cancer types	62.7 ± 12.5	58.4	10	n/a	3.0 (0.0–74.4)	+	+
2	Braun, 2011 [20]	Cohort	1194	Stage I–IV NSCLC	58.5 (21.6–86.4)	50.3	n/a	65.2	8.8 (8.0–9.5)	+	-
3	Brunello, 2016 [21]	Cohort	658	Stage I–IV; various cancer types	77.2 ± 5.1	34.2	12	17.4	n/a	+	+
4	Cesari, 2013 [22]	Cohort	200	Stage I–IV ovarian or uterine cancer	73.5 ± 6.2	0.0	12	11.5	n/a	+	+
5	Chen, 2019 [23]	Cohort	121	Gastric adenocarcinoma; stage I–IV	64.0 ± 14.9	60.3	12	32.2	n/a	-	-
6	Chow, 2008 [24]	Cohort	395	Metastatic disease, referred for palliative radiotherapy; various cancer types	68.0 (31.0–93.0)	50.0	12	n/a	4.4 (3.9–6.4)	+	+
7	Collette, 2004 [25]	RCT	391	Metastatic hormone-refractory prostate cancer	70.8 (34.3–89.3)	100.0	12	42.7	10.4 (9.2–11.5)	+	+
8	Collins, 2014 [26]	Cohort	1160	Malignant glioma	n/a	58.4	4	23.0	n/a	-	-
9	Contreras-Bolívar, 2019 [27]	Cohort	282	Various cancer types	60.4 ± 12.6	55.7	6	47.9	n/a	-	-
10	Deans, 2007 [28]	Cohort	220	Stage I–IV gastric or esophageal cancer	71.0 (62.0–78.0)	65.9	24	61.8	13.0 (n/a)	+	+
11	Dharma-Wardene, 2004 [29]	Cohort	42	Stage IIIA/B or IV NSCLC or SCLC	59.4 (39.0–78.0)	45.2	24	16.7	9.9 (n/a)	+	-
12	Efficace, 2006 [30]	RCT	391	Stage IIIB–IV NSCLC	57.0 (28.1–75.9)	65.2	n/a	77.2	(6.7–8.9)	+	-
13	Ferrigno, 2001 [31]	Cohort	343	Stage 0–IV bronchogenic carcinoma	68.0 (39.0–86.0)	88.6	8.5 (3.6–17.4)	79.0	n/a	-	-
14	Fielding, 2007 [32]	Cohort	358	Lung (analysis, stage I–IV)	n/a, subgroups of different ages available	75.7	3.2 (2.0–7.0)	78.5	n/a	+	+
15	Filippini, 2008 [33]	Cohort	676	Newly-diagnosed glioblastoma	58.0 (16.0–81.0)	61.8	24	84.0	13.6 (12.9–14.3)	+	-
16	Gagnon, 2013 [34]	Cohort	258	Inoperable stage III–IV NSCLC	n/a	50.0	12	n/a	(2.5–18.2)	-	+
17	Geraci, 2006 [35]	Cohort	372	Acutely symptomatic cancer patients; various cancer types	56.0 (15.0–96.0)	49.0	6	30.0	n/a	-	-
18	Giantin, 2013 [36]	Cohort	160	Inoperable, locally advanced or metastatic cancer; various cancer types	79.4 ± 5.7	45.5	12	46.9	n/a	+	+
19	Griguolo, 2018 [37]	Cohort	668	Invasive breast cancer, brain metastases	56.0 (24.0–85.0)	0.1	n/a	94.6	8.1 (6.9–9.4)	+	-
20	Gripp, 2007 [38]	Cohort	216	Patients examined for palliative radiation; various cancer types	64.0 (21.0–96.0)	51.0	6	51.4	n/a	-	-
21	Gupta, 2004 [39]	Cohort	58	Stage IV pancreatic cancer	56.2 ± 10.7	60.3	n/a	72.4	(4.9–10.2)	-	-
22	Gupta, 2009 [40]	Cohort	165	Stage IIIB or IV NSCLC	56.0 (30.0–78.0)	56.4	n/a	67.3	(6.8–16.8)	-	-
23	Hoang, 2005 [41]	RCT	1436	Stage IIIB or IV NSCLC	n/a	63.0	24	89.0	8.2 (n/a)	+	+
24	Hong, 2016 [42]	Cohort	183	Advanced HCC	Mean 55.8	86.3	6	74.9	n/a	-	+
25	Hosono, 2005 [43]	Cohort	165	Spinal metastases; various cancer types	n/a	n/a	23.4 (0.3–140.0)	n/a	n/a	-	-
26	Hui, 2014 [44]	Cohort	222	Advanced cancer, seen by palliative care mobile team; various cancer types	55.0 (22.0–79.0)	41.0	3.9 (0.9–7.9)	64.0	3.6 (2.3–4.2)	+	-
27	Hui, 2016 [45]	Cohort	216	Advanced cancer, seen by the palliative care mobile team; various cancer types	54.9 (22.0–79.0)	42.0	7.9 (6.9–8.6)	63.0	3.6 (2.3–4.4)	-	+
28	Iversen, 2009 [46]	Cohort	13,190	Stage I–IV colorectal cancer	n/a	47.0	12	n/a	n/a	+	-
29	Jang, 2014 [47]	Cohort	1655	Advanced cancer; various cancer types	65.0 (21.0–97.0)	49.0	n/a	91.0	4.4 (4.0–4.7)	-	+
30	Jonna, 2016 [48]	Cohort	803	Stage I–IV; various cancer types	72.5 (n/a)	51.8	12	77.3	4.9 (n/a)	+	+
31	Katagiri, 2005 [49]	Cohort	350	Bone metastases; various cancer types	59.0 (14.0–88.0)	56.9	24	67.0	n/a	+	-
32	Kilgour, 2013 [50]	Cohort	203	Locally, advanced and metastatic cancer; various cancer types	64.3 ± 12.8	57.1	n/a	76.6	7.3 (5.3–9.3)	-	-
33	Kim, 2009 [51]	Cohort	325	UICC stage I–IV newly-diagnosed HCC	58.8 ± 9.5	80.9	24	46.4	14.7 (2.0–88.0)	-	+
34	Kinoshita, 2012 [52]	Cohort	133	Stage I–IV HCC	71.0 (43.0–87.0)	70.7	22.0 (1.0–69.0)	13.0–91.4	n/a	+	-
35	Langendijk, 2000 [53]	Cohort	198	Stage I–IIIB NSCLC	n/a	85.0	n/a	n/a	(2.8–13.8)	-	-
36	Liljehult, 2017 [54]	Cohort	109	Glioblastoma	65.0 ± 9.9	56.9	12	55.0	n/a	-	-
37	Limquiaco, 2009 [55]	Cohort	471	Stage I–IV HCC	58.8 ± 12.2	85.1	6	45.0	10.1 ± 10.3	-	+
38	Lund, 2009 [56]	Cohort	2315	Renal cancer	Men: 68.0 (15.0–96.0); women: 70.0 (18.0–97.0)	58.7	12	36.9	n/a	+	-
39	Maione, 2005 [57]	RCT	566	Stage IIIB–IV NSCLC	74.0 (70.0–84.0)	82.0	12	68	6.9 (6.4–7.8)	+	-
40	Marrero, 2005 [58]	Cohort	244	Stage I–IV HCC	57.0 ± 10.0	73.0	12	42.0	16.4 (12.9–19.8)	+	-
41	Martin, 2010 [59]	Cohort	1164	Metastatic cancer; various cancer types	66.8 ± 13.0	49.0	3.1 (0.0–38.6)	86.4	n/a	+	+
42	Moroni, 2014 [60]	Cohort	231	Advanced cancer; various cancer types	70.2 ± 0.9	50.6	12	45.0	n/a	+	+
43	Moss, 2010 [61]	Cohort	826	Stage I–IV breast, lung, and colon cancer	60.0 ± 13.0	14.8	12	8.3	n/a	+	+
44	Motzer, 2004 [62]	Cohort	251	Stage IV renal cell carcinoma	57.0 (23.0–77.0)	67.0	24	76.0	10.2 (8.0–12.0)	+	-
45	Norman, 2010 [63]	Cohort	399	Stage I–IV; various cancer types	63.0 ± 11.8	52.1	6	25.1	n/a	+	-
46	Orskov, 2016 [64]	Cohort	2654	Stage I–IV ovary cancer	n/a	0.0	12	16.0	n/a	-	-
47	Park, 2016 [65]	Cohort	403	Metastatic or recurrent pancreatic ductal adenocarcinoma	66.0 (29.0–96.0)	49.1	7.9 (0.1–70.5)	n/a	8.2 (7.3–9.1)	+	+
48	Penel, 2008 [66]	Cohort	119	Bone metastases; various cancer types	57.0 (29.0–84.0)	64.7	3	34.0	3.9 (0.0–94.5)	+	+
49	Penel, 2008 [67]	Cohort	148	Patients screened for phase 1 trial; various cancer types	54.0 (23.0–79.0)	n/a	3	73.0	5.7 (0.0–79.6)	+	-
50	Pinato, 2015 [68]	Cohort	97	Intermediate-advanced HCC	64.0 (22.0–82.0)	80.0	n/a	71.0	5.7 (1.0–88.0)	-	+
51	Pointillart, 2011 [69]	Cohort	142	Vertebral metastases; various cancer types	61.8 (28.0–89.0)	57.0	12	50.7	5.0 (n/a)	+	-
52	Roychowdhury, 2003 [70]	Cohort	364	Locally advanced or metastatic urothelial transitional-cell carcinoma	63.5 (n/a)	79.1	n/a	n/a	14.2 (13.1–16.8)	-	-
53	Rydzek, 2015 [71]	Cohort	326	Breast cancer, SCLC or NSCLC; patients with established cardiovascular disease	67.8 ± 10.0	54.0	12	n/a	(4.8–96.0)	+	-
54	Schoenfeld, 2020 [72]	Cohort	1216	Various cancer types, spinal metastases	58.0 ± 9.7	50.0	12	47.0	8.4 (3.1–21.1)	-	-
55	Scott, 2002 [73]	Cohort	106	Stage III–IV NSCLC	69.0 (43.0–87.0)	58.5	n/a	n/a	5.2 (0.3–38.5)	+	-
56	Seow, 2013 [74]	Cohort	11,342	Various cancer types, patients with PPS assessment	64.0 ± n/a	45.6	6	25	n/a	+	-
57	Shen, 2007 [75]	Cohort	49	HCC, patients underwent TACE	57.0 ± 1.0	81.6	n/a	49.0	12.0 (1.0–72.0)	-	-
58	Soubeyran, 2012 [76]	Cohort	348	Various cancer types; 65% advanced stage	77.5 (70.0–99.4)	59.5	6	16.1	n/a	+	-
59	Sutradhar, 2014 [77]	Cohort	66,112	Various cancer types	n/a	43.8	19.3 (n/a)	26.3	n/a	-	+
60	Suzuki, 2020 [78]	Cohort	185	Advanced urothelial cancer	70.0 (64.0–76.0)	68.0	12	71.9	14.9 (n/a)	-	-
61	Tsai, 2014 [79]	Cohort	522	Advanced cancer; various cancer types	60.6 ± 13.2	61.7	6	91.8	n/a	-	-
62	Tripodoro, 2019 [80]	Cohort	317	Various cancer types, stage III or IV	77.0 (21.0–99.0)	33.4	24	74.8	4.0 (n/a)	-	-
63	Ueno, 2000 [81]	Cohort	103	Metastatic pancreatic cancer	62.0 (42.0–75.0)	68.0	12	94.4	3.2 (n/a)	+	-
64	van der Linden, 2005 [82]	RCT	342	Spinal metastases; various cancer types	66.0 (34.0–90.0)	53.0	24	75.0	11.0 (10.0–12.0)	-	-
65	Vigano, 2000 [83]	Cohort	227	Inoperable, recurrent, progressive or metastatic cancer; various cancer types	62.0 (29.0–92.0)	36.1	20	91.6	5.8 (n/a)	+	-
66	Villa, 2011 [84]	Cohort	285	Newly-diagnosed brain metastases; various cancer types	62.0 (20.0–90.0)	n/a	12	82.4	n/a	+	+
67	Wei, 2019 [85]	Cohort	71	Pneumonic-type adenocarcinoma, 90% with stage IIIB or IV	62.0 (25.0–91.0)	45.1	6	84.5	7.5 (1.0–42.0)	+	-
68	Yamashita, 2011 [86]	Cohort	85	Spinal metastases; various cancer types	60.3 ± 11.6	51.8	12	51.8	n/a	-	-

H: high risk-of-bias; HCC: hepatocellular carcinoma; L: low risk-of-bias; M: moderate risk-of-bias; N: number of patients; n/a: not available; NSCLC: non-small cell lung cancer; PPS: Palliative Performance Scale; RCT: randomized controlled trial; SCLC: small cell lung cancer; TACE: trans arterial chemoembolization; UICC: Union Internationale Contre le Cancer.

**Table 2 cancers-14-00328-t002:** Summary of risk-of-bias assessment.

Study	Study Participation	Study Attrition	Predictors	Outcome	Statistical Analysis and Cofounding	Performance of Prediction Tool	Overall Bias
Bartels, 2007 [19]	L	H	M	L	L	M	M
Braun, 2011 [20]	L	H	L	L	L	n/a	L
Brunello, 2016 [21]	L	H	L	L	M	M	M
Cesari, 2013 [22]	L	H	L	L	M	M	M
Chen, 2019 [23]	L	M	L	L	H	n/a	M
Chow, 2008 [24]	L	H	L	M	M	L	M
Collette, 2004 [25]	M	H	L	L	L	M	M
Collins, 2014 [26]	L	M	L	L	L	n/a	L
Contreras-Bolívar, 2019 [27]	L	M	L	L	L	n/a	L
Deans, 2007 [28]	L	M	L	L	L	M	L
Dharma-Wardene, 2004 [29]	L	H	L	L	M	n/a	M
Efficace, 2006 [30]	L	H	M	L	L	n/a	M
Ferrigno, 2001 [31]	L	H	L	L	L	n/a	L
Fielding, 2007 [32]	L	H	L	L	L	M	M
Filippini, 2008 [33]	L	M	M	L	L	n/a	L
Gagnon, 2013 [34]	L	H	L	M	L	L	M
Geraci, 2006 [35]	L	H	L	L	L	n/a	L
Giantin, 2013 [36]	L	H	L	L	L	M	M
Griguolo, 2018 [37]	L	H	L	L	L	n/a	L
Gripp, 2007 [38]	L	H	L	L	L	n/a	L
Gupta, 2004 [39]	L	H	L	L	L	n/a	L
Gupta, 2009 [40]	L	M	L	L	M	n/a	L
Hoang, 2005 [41]	L	H	M	L	L	H	H
Hong, 2016 [42]	L	H	L	L	M	M	M
Hosono, 2005 [43]	M	H	M	L	M	n/a	H
Hui, 2014 [44]	L	H	L	L	L	n/a	L
Hui, 2016 [45]	L	H	L	L	H	M	H
Iversen, 2009 [46]	L	H	L	L	M	n/a	M
Jang, 2014 [47]	L	H	L	L	H	M	H
Jonna, 2016 [48]	L	H	M	L	L	M	M
Katagiri, 2005 [49]	L	H	M	L	M	n/a	M
Kilgour, 2013 [50]	L	L	L	L	M	n/a	L
Kim, 2009 [51]	L	H	M	L	M	M	H
Kinoshita, 2012 [52]	L	H	L	L	M	n/a	M
Langendijk, 2000 [53]	L	H	L	M	M	n/a	M
Liljehult, 2017 [54]	M	H	L	L	M	n/a	M
Limquiaco, 2009 [55]	L	H	L	L	L	M	M
Lund, 2009 [56]	L	M	L	L	H	n/a	M
Maione, 2005 [57]	L	H	L	L	M	n/a	M
Marrero, 2005 [58]	L	M	M	L	M	n/a	M
Martin, 2010 [59]	L	H	L	L	L	L	L
Moroni, 2014 [60]	L	H	L	L	L	M	M
Moss, 2010 [61]	L	H	L	L	L	M	M
Motzer, 2004 [62]	L	H	L	L	L	n/a	L
Norman, 2010 [63]	L	H	L	L	M	n/a	M
Orskov, 2016 [64]	L	M	L	L	M	n/a	L
Park, 2016 [65]	L	H	L	M	L	M	M
Penel, 2008 [66]	L	H	M	L	M	M	H
Penel, 2008 [67]	L	H	L	L	M	n/a	M
Pinato, 2015 [68]	L	H	M	L	L	L	M
Pointillart, 2011 [69]	L	H	L	L	M	n/a	M
Roychowdhury, 2003 [70]	L	H	L	M	M	n/a	M
Rydzek, 2015 [71]	L	H	L	M	M	n/a	M
Schoenfeld, 2020 [72]	L	H	L	L	L	n/a	L
Scott, 2002 [73]	L	L	L	L	L	n/a	L
Seow, 2013 [74]	L	H	L	L	M	n/a	M
Shen, 2007 [75]	L	H	L	L	L	n/a	L
Soubeyran, 2012 [76]	L	H	L	L	H	n/a	M
Sutradhar, 2014 [77]	L	H	M	L	L	M	M
Suzuki, 2020 [78]	L	H	M	L	L	n/a	M
Tripodoro, 2019 [80]	L	H	L	L	M	n/a	M
Tsai, 2014 [79]	L	M	L	L	L	n/a	L
Ueno, 2000 [81]	L	H	L	L	L	n/a	L
van der Linden, 2005 [82]	L	M	L	L	M	n/a	L
Vigano, 2000 [83]	L	M	L	L	L	n/a	L
Villa, 2011 [84]	L	M	M	L	L	M	M
Wei, 2019 [85]	L	M	M	M	M	n/a	M
Yamashita, 2011 [86]	L	H	L	L	L	n/a	L

H: high risk-of-bias; L: low risk-of-bias; M: moderate risk-of-bias; n/a: not applicable.

**Table 3 cancers-14-00328-t003:** Summary of the prediction models.

				Variables																		
Study	FU	N	Name Model	Cancer Type	Performance Status	Cancer Treatment	Sex	Laboratory Results	Metastases	Disease Stage	Age	Cognitive Impairment	Dietary Intake	Weight Change	Body Mass Index	Surprise Question	Comorbidity	(Re)hospitalization	Other	Discriminative Ability	Calibration	External Validation
Bartels, 2007 [19]	10	219	n/a	*	*	*	*		*											c-statistic 0.72 (95% CI 0.68–0.77)	n/a	n/a
Brunello, 2016 [21]	12	658	Onco-MPI	*	* *		*			*	*	*			*				* * *	c-statistic 0.869 (95% CI 0.841–0.897)	Good	n/a
Cesari [22]	12	200	IADL																*	AUC 0.676 (95% CI 0.532–0.821)	n/a	n/a
			SPPB																*	AUC 0.638 (95% CI 0.483–0.792)	n/a	n/a
			UGS																*	AUC 0.686 (95% CI 0.560–0.812)	n/a	n/a
Chow, 2008 [24]	12	395	n/a	*	*				*											c-statistic 0.66	n/a	*N* = 467 c-statistic 0.63
Deans, 2007 [28]	24	220	u/a		*			*		*				*						AUC 0.85	n/a	n/a
Giantin, 2013 [36]	12	160	MPI	*		*						*							* * *	AUC 0.874 (95% CI 0.819–0.928)	n/a	n/a
Jang, 2014 [47]	n/a	1655	KPS		*															c-statistic 0.63	n/a	n/a
			PPS		*															c-statistic 0.63	n/a	n/a
			ECOG		*															c-statistic 0.64	n/a	n/a
Jonna, 2016 [48]	12	803	n/a	*		*	*					*			*			* *	*	c-statistic 0.66 (95% CI 0.58–0.72)	n/a	n/a
Martin, 2010 [59]	3.1	1164	PG-SGA	*	*								*	*					*	c-statistics 0.88 (95% CI 0.83–0.91)	n/a	*N* = 603 c-statistic 0.87 (95% CI 0.80–0.92)
Moroni, 2014 [60]	12	231	Surprise question													*				Se 69.3 (95% CI 60.5–77.2)/ Sp 83.6 (95% CI 75.1–90.2)/ PPV = 83.8 (95% CI 75.3–90.3)/ NPV = 69.0 (95% CI 60.2–77.0)	n/a	n/a
Moss, 2010 [61]	12	826	Surprise question													*				Se 75/Sp 90/PPV 41/NPV 97	n/a	n/a
Penel, 2008 [66]	3	119			*			* * *	* * *											Se 0.21–0.89/Sp 0.33–0.96/ Positive LR 0.40–0.78/ Negative LR 0.72–0.85/	n/a	n/a
Sutradhar, 2014 [77]	19.3	66,112	n/a	*			*				*						*		*	c-statistic 0.764–0.872 (95% CI 0.758–0.878)	n/a	n/a

ECOG: Eastern Cooperative Oncology Group; FU: follow-up; KPS: IADL: instrumental activities of daily living; Karnofsky Performance Status; LR: likelihood ratio; MPI: multidimensional prognostic index; n/a: not available; NPV: negative predictive value; Onco-MPI: Oncological-multidimensional prognostic index; PG-SGA: Patient-generated Subjective Global Assessment; PPS: Palliative Performance Scale; PPV: positive predictive value; Se: sensitivity; Sp: specificity; SPPB: Short Physical Performance Battery; UGS: usual gait speed. Legend: an asterisk represents the presence of the variable, whereby two or more asterisks per variable represent different subcategories.

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
