# Peer review of "Predictors of Mortality in Patients with Advanced Cancer—A Systematic Review and Meta-Analysis"

_cancers, 2022, doi:10.3390/cancers14020328_

Round 1

Reviewer 1 Report

I previously reviewed this manuscript. All of my comments have been addressed. I have no further comments.

Reviewer 2 Report

1. Abstract, line 24 , “we included 69 studies (42-66, 112 patients)…”

I am confused with the patient number. Based on table 1, Braun’s study (ref 20) was included in your meta analysis and 1194 patients were recruited in Braun’s cohort. Therefore, I estimated the patients number was much more than 112 in this meta analysis.

2. Method, line 83, Please identify “had a combined outcome”

3. Included studies were published from 2004-2019. It may be necessary to adjust the published year in order to decrease the impact from the medical progress in the past decades. 

4. Ref 37 & Ref38 were implemented by the same group in the same institution for the same population (breast cancer with brain metastasis). How to make sure no duplicated patients were analyzed in ref 37 and ref 38.

5. Ref 32 was included in the meta analysis for age v.s. mortality (figure 3), however, the information about age was N/A in table 1

Round 2

Reviewer 2 Report

Good response,  and now the current manuscript is suitable to be published in Cancers 

This manuscript is a resubmission of an earlier submission. The following is a list of the peer review reports and author responses from that submission.

Round 1

Reviewer 1 Report

The manuscript “Predictors of mortality in patients with advanced cancer – A systematic review and meta-analysis” aimed to investigate predictors and prediction models for mortality within a period of 3-24 months for patients with advanced cancer.

Overall, the manuscript is well written in a clear and concise manner.

  • A study protocol was published at PROSPERO.
  • The aim of the study is clearly defined.
  • The study question is clinically relevant and focused.
  • The authors performed a comprehensive literature search including several databases (Embase, MEDLINE, Cochrane Central, Web of Science, and PubMed), and screened the reference lists of included studies as is appropriate. Also, the authors did search for grey literature and requested full articles that were unavailable.
  • The researchers adequately provided the terms that they used for electronic literature search.
  • The inclusion and exclusion criteria used for selection of studies are clearly stated.
  • Characteristics of studies that were included in the analysis are listed in full detail.
  • The procedures for solving disagreement during studies’ inclusion as well as data extraction were explained in detail.
  • The risk-of-bias of individual studies was examined using the Quality in Prognosis Studies with adapted items from the Critical Appraisal and Data Extraction for Systematic Reviews of Prediction Modelling Studies checklist.
  • Assessment of publication bias was done using a funnel plot.
  • The flowchart with number of included and excluded studies, along with reasons for their exclusion, was adequately presented.
  • The authors stated limitations of their work, as is necessary.
  • The authors provided enough detail to enable a replication of this work.

Reviewer 2 Report

  1. The literature search are up to April 2020. Can you update the literature search.
  2. There were 69 studies included in qualitative synthesis, but 40 studies included in meta-analysis. Please clarify what is qualitative synthesis
  3. “The overall risk-of-bias (low, moderate, or high) of each study was generated from the points given to the items per domain” Please specify the cut-off point for low, moderate, high risk-of-bias for overall risk
  4. “The risk estimates and standard errors of predictors that were significant in multivariable analyses in published studies, were included in the meta-analysis” why did you included only predictors that were significant in multivariable analyses in the meta-analysis.
  5. “Table 1. Study characteristics of all 64 studies reporting predictors of mortality or prediction models” should it be 69 studies
  6. In table 1, several studies include stage I cancer. Is stage I cancer considered advanced cancer?
  7. C-statistic and AUC are the same. If so, please use consistent word in table 3
  8. Why did you not perform meta-analysis of c-statistic of prediction model

Reviewer 3 Report

Major comments

1. Some studies have short follow-up periods and low mortality rates, but do they affect results?

2.     Did you test for heterogeneity?Minor comments1.     Figure S1overlap and can’t be printed accurately.

Round 2

Reviewer 2 Report

all of my comments have been addressed 

but the literature search was done in April 2020. This is about 1.5 years ago. I suggest them to update the literature search but they refused but would do it if the editor suggest. 

Reviewer 3 Report

The conclusion is not supported by the result.